# The Conjunctive Compensation Method Based on Inertial Navigation System and Fluxgate Magnetometer

Bingyang Chen [1,2,3], Ke Zhang [1,2,3], Bin Yan [1,2] and Wanhua Zhu [1,2,*]

1. Aerospace Information Research Institute, Chinese Academy of Sciences, Beijing 100190, China; 15291172@bjtu.edu.cn (B.C.)
2. Key Laboratory of Electromagnetic Radiation and Sensing Technology, Chinese Academy of Sciences, Beijing 100190, China
3. School of Electronic, Electrical and Communication Engineering, University of Chinese Academy of Sciences, Beijing 100190, China
* Correspondence: whzhu@mail.ie.ac.cn

**Abstract:** Eliminating the magnetic interference of the carrier platform is an important technical link and plays a vital role in aeromagnetic survey. The traditional compensation method is based on the Tolles–Lawson (T-L) model and establishes the linear relationship between the aircraft interference magnetic field and the aircraft attitude. The compensation coefficients are solved by designing the calibration flight. At present, almost all aeromagnetic systems use the fluxgate magnetometer fixed to the aircraft to realize the attitude measurement of the flight platform. However, the fluxgate magnetometer has problems, such as non-orthogonal error, zero drift error, and linearity error limited by the production process, and the fluxgate magnetometer is also very susceptible to external magnetic interference as a magnetic sensor. These lead to the aircraft attitude calculated by the fluxgate magnetometer being inaccurate, thus reducing the compensation effect. In this article, we analyze the influence of the fluxgate magnetometer noise on compensation and propose a new conjunctive compensation method based on inertial navigation systems (INS) and fluxgate magnetometer information to improve the compensation effect. The flight experiment data show that the proposed method can significantly improve the quality of aeromagnetic data. Compared with the traditional compensation method only based on fluxgate magnetometer information, the improved ratio is increased by 30–60%, and it is a real-time compensation method. It shows that the proposed method has a remarkable compensation effect for aeromagnetic interference.

**Keywords:** aeromagnetic compensation; inertial navigation system; conjunctive compensation

## 1. Introduction

Aeromagnetic survey is a method to detect weak magnetic anomaly signals of underground or underwater targets by using an aircraft equipped with high-precision magnetometer to scan the doubtful areas [1]. Comparing with ground magnetic survey, it has the advantages of high detection efficiency and large operation area [2], so it is widely used in geophysical exploration [3], mineral exploration [4], and underwater anti-submarine detection [5].

Helicopters and fixed-wing aircrafts are the most commonly used carrier platforms in aeromagnetic survey. In the process of aeromagnetic measurement, the ferromagnetic materials and conductive materials of the aircraft platform can produce interference magnetic fields when the platform is maneuvering [6]. It affects the accuracy of the magnetometer and limits its detection performance. Therefore, eliminating the magnetic interference of the aircraft platform plays a vital role in aeromagnetic measurement and is an important technical link of aeromagnetic surveying. In 1950, Tolles and Lawson proposed that the aircraft's interference field can be divided into three parts: (1) remanent magnetic field;

(2) induced magnetic field; and (3) eddy current field [7–9], and established a linear compensation model with 18 coefficients related to magnetic interference and attitude of aircraft. The accuracy and stability of model coefficients determine the compensation effect. The compensation coefficient was calculated by calibration flight. In order to improve the accuracy of coefficient calculation, it is necessary to obtain the attitude information of the aircraft as accurately as possible.

The sensor that can directly reflect the change in aircraft attitude is INS, but it is not clear how to substitute the INS information into a T-L model and the INS is not precise enough. The current mainstream aeromagnetic compensation system mainly consists of an optical pump magnetometer (OPM) and fluxgate magnetometer. An OPM is used to measure the field with high precision. Fluxgate magnetometers measure magnetic field vectors. The fluxgate magnetometer is fixed on the aircraft platform, and the change in aircraft attitude can be obtained directly through the measured values of the three axes of the fluxgate magnetometer and applied to the T-L model. However, due to the limitation of production technology and processing level, the fluxgate magnetometer has some problems, such as non-orthogonal error [10], zero drift error [11], and temperature drift [12–14], and as a magnetic sensor, a fluxgate magnetometer is also easily affected by external magnetic interference. As a result, the change in fluxgate magnetic field value caused by aircraft maneuvering may be submerged, thus making the compensation coefficient obtained by calibrated flight inaccurate, thus reducing the compensation effect. In terms of fluxgate error correction, J. Liu proposed other sensors, such as an inertial measurement unit (IMU) or gyro, also used to calibrate magnetic sensors [15]. In [16], the author proposed a fluxgate magnetometer calibration method by using the least square method and particle swarm optimization algorithm. Zhuo Chen introduced a hybrid optimizing algorithm to compensate the dynamic error of fluxgate magnetometers [17]. However, due to complex working conditions and many interference factors, equipment errors are often unpredictable, and the influence mechanism of fluxgate interference on compensation is not clear. This makes it difficult to improve the compensation effect by adjusting the fluxgate magnetometer.

In order to reduce the influence of fluxgate error on aeromagnetic compensation and improve the compensation accuracy, this paper analyzes the influence of the fluxgate magnetometer noise on compensation and the simulation experiment is carried out. The simulation results show that the larger the noise of the fluxgate magnetometer, the worse the compensation results. With reference to the T-L model modeling method, this paper proposes a new conjunctive compensation method based on INS/GPS and fluxgate magnetometer information. Through modeling, inertial navigation information is introduced into the T-L model, and the compensation model with 18 coefficients is transformed to the compensation model with 36 coefficients. The negative effect of the fluxgate error in the compensation model is suppressed. The flight experiment data show that the proposed method can significantly improve the quality of aeromagnetic data and it is also a real-time compensation method. Compared with the traditional compensation method only based on fluxgate magnetometer information, the improved ratio is increased significantly.

This paper briefly introduces the T-L model algorithm and analyzes theoretically the influences of fluxgate noise on compensation. Then, the influence of different amplitude fluxgate noise on the compensation results is compared by experimental flight data. Finally, the conjunctive compensation method based on INS/GPS is described in detail and the compensation results and analysis are given.

## 2. Related Work

### 2.1. T-L Model

The airborne magnetic survey systems consist of multiple modules that are mainly composed of OPM, fluxgate magnetometer, radar altimeter, INS, GPS and other units. Among them, OPM is a scalar total field magnetometer, which realizes high-precision measurement of the magnetic field [18]. Its measurement is the sum of the modulus of the

geomagnetic field vector and other magnetic field vectors (magnetic interference, target signal, etc.). It is equivalent numerically to the sum of the modulus of the geomagnetic field vector and the projections of other magnetic fields in the direction of the geomagnetic field. The obtained signal is the superposition of other interference fields and magnetic target signals after filtering the slowly changing geomagnetic field. So, it is important to eliminate the magnetic interference of the aircraft platform for aeromagnetic measurement. Fluxgate magnetometer can measure the vector of the magnetic field. It can obtain the information of the angle between the coordinate system of the flight platform and the geomagnetic field vector through the fluxgate magnetometer fixedly connecting to the flight platform, so as to measure the attitude of the flight platform. A radar altimeter uses radar echo to measure the altitude of the flight platform to the ground. The INS records the angle of roll, pitch, and yaw speed and other motion parameter information of the aircraft. The GPS records the longitude, latitude, and altitude information of the flight platform and provides auxiliary information for data post-processing.

The space of the aircraft is sufficient, so the magnetometer can be installed far away from the magnetic interference sources, such as electronic equipment, to eliminate the magnetic interference of electronic equipment. Therefore, the magnetic interferences caused by the interaction between the airframe material and the background geomagnetic field during the aircraft movement are predominant. In 1950, Tolles and Lawson proposed that the aircraft interference field can be divided into three parts: permanent magnetic field, induction field, and eddy current field, and established a compensation model with 18 coefficients [19]:

$$H_I = H_{IP} + H_{II} + H_{IE} = \sum_{i=1}^{3} p_i c_i + H_f \sum_{i=1}^{3} \sum_{j=1}^{3} a_{ij} c_i c_j + H_f \sum_{i=1}^{3} \sum_{j=1}^{3} b_{ij} \dot{c}_i c_j \tag{1}$$

where $p_i, a_{ij}$, and $b_{ij}$ are the coefficient of the T-L model, $H_f$ is the total magnetic field intensity measured by fluxgate magnetometer, $c_i$ is the cosine of the angle between the three axes of the aircraft coordinate system, which can be measured from the three axes of the fluxgate magnetometer $H_i$, $i = 1, 2, 3$, the expression is (2), and $\dot{c}_i$ is time derivative of $c_i$

$$c_i = \frac{H_i}{\sqrt{H_1^2 + H_2^2 + H_3^2}}, i = 1, 2, 3. \tag{2}$$

Then, the measured value of OPM can be expressed as the superposition of the geomagnetic field and aircraft magnetic interference as follows:

$$H_M = H_I + H_E \tag{3}$$

where $H_E$ is the geomagnetic field and $H_M$ is the measured value of OPM. Generally, the earth's magnetic field is slowly changing and its frequency is extremely low. Therefore, the low-frequency geomagnetic field and high-frequency interference can be eliminated by the suitable band-pass filter. After band-pass filtering, the magnetic data only contain magnetic anomaly signals and aircraft magnetic interference.

In the detection of aeromagnetic targets, the magnetic anomaly signals generated by the detected targets are usually in the extremely low frequency band (<1 Hz). In order to match the target recognition, we usually pay attention to the low-frequency magnetic field with the frequency range of 0.04–0.3 Hz. In this paper, the band of the band-pass filter is set to 0.04–0.3 Hz.

N data points are obtained by calibration flight, and then the T-L linear model can be expressed as matrix follows:

$$B = AC + z \tag{4}$$

$B$ represents n scalar magnetometer field data after preprocessing. A is $n \times 18$ platform attitude matrix. $C$ is $18 \times 1$ coefficients matrix of T-L compensation model. It is important

to solve the compensation coefficients by calibration flight. Therefore, calibration flight is needed before aeromagnetic survey. The least square solution of the compensation coefficients $C$ are as follows [20]:

$$C = \left(A^T A\right)^{-1} A^T B. \tag{5}$$

In order to obtain good calibration flight data, calibration flight needs some certain requirements. Firstly, the region with a stable geomagnetic field is usually selected for calibration flight. A complete compensation flight needs to be performed in four directions, and three sets of actions need to be completed in each direction: pitch, yaw, and roll. Each action needs to be repeated at least three times, where the standard amplitude of pitch flight is $\pm 5^{\circ}$, the standard amplitude of yaw flight is $\pm 10^{\circ}$, and the standard amplitude of roll flight is $\pm 5^{\circ}$. After completing the calibration flight, 18 T-L compensation coefficients can be obtained by solving the T-L linear equation.

After the compensation is completed, the compensation results should be evaluated. The compensation results should be evaluated after the compensation is completed. At present, the general evaluation standard of compensation results of the aeromagnetic survey is to use standard deviation (STD) as the measure of data interference, and to use improved ratio (IR) to measure the compensation results of the algorithm on data [21]. The higher the IR, the better the compensation results. STD is defined as (6) and IR as (7).

$$STD = \sqrt{\frac{1}{n} \sum_{i=1}^{n} (x_i - \mu)} \tag{6}$$

$$IR = \frac{STD(H_b)}{STD(H_a)} \tag{7}$$

where $\mu$ is the arithmetic mean of $x_i$ and $H_b$, $H_a$ represents the OPM data before and after compensation, respectively.

### 2.2. The Compensation Influence of the Fluxgate Magnetometer Noise

In aeromagnetic compensation, the T-L compensation model is as follows:

$$H_I = H_{IP} + H_{II} + H_{IE} \tag{8}$$

$$H_{IP} = c_1 \frac{T}{H_e} + c_2 \frac{L}{H_e} + c_3 \frac{V}{H_e} \tag{9}$$

$$H_{II} = \left(c_4 \frac{T^2}{H_e} + c_5 \frac{L^2}{H_e} + c_6 \frac{V^2}{H_e} + c_7 \frac{TL}{H_e} + c_8 \frac{TV}{H_e} + c_9 \frac{LV}{H_e}\right) \tag{10}$$

$$\begin{aligned} H_{IE} = &\, c_{10} T\left(\frac{T}{H_e}\right)' + c_{11} L\left(\frac{L}{H_e}\right)' + c_{12} V\left(\frac{V}{H_e}\right)' + c_{13} T\left(\frac{L}{H_e}\right)' + c_{14} T\left(\frac{V}{H_e}\right)' + \\ &\, c_{15} L\left(\frac{V}{H_e}\right)' + c_{16} L\left(\frac{T}{H_e}\right)' + c_{17} V\left(\frac{T}{H_e}\right)' + c_{18} V\left(\frac{L}{H_e}\right)' \end{aligned} \tag{11}$$

$$H_e = \sqrt{T^2 + L^2 + V^2} \tag{12}$$

where $H_{IP}$ is permanent magnetic field, $H_{II}$ is induction field, and $H_{IE}$ is the eddy current field. $H_e$ is the total magnetic field measured by the fluxgate magnetometer, and $c_i$ is the coefficient of the T-L model. $T$, $L$, and $V$ represent the true value of the magnetic field on the three axes X, Y, and Z of the fluxgate magnetometer.

The magnetic interference of the aircraft platform is a quantity $H_I$ $(T, L, V)$ related to the magnetic field value on the three-axis of the fluxgate. When the magnetic field value

on the three-axis deviates from the real value, the deviation of the aircraft interference magnetic field can be expressed as follows:

$$\Delta H_I = H_I(T + \Delta T, L + \Delta L, V + \Delta V) - H_I(T, L, V). \tag{13}$$

where $T$, $L$, and $V$ are the true values of the magnetic field on the three axes of the fluxgate magnetometer while $\Delta T$, $\Delta L$, and $\Delta V$ represent the deviations between the measured values and the true values on the three axes of the fluxgate magnetometer, respectively. Taylor expansion of $H_I$ $(T + \Delta T, L + \Delta L, V + \Delta V)$ yields the following:

$$H_I(T + \Delta T, L + \Delta L, V + \Delta V) = H_I(T, L, V) + \frac{\partial H_I}{\partial T}\Delta T + \frac{\partial H_I}{\partial L}\Delta L + \frac{\partial H_I}{\partial V}\Delta V + o(T, L, V) \tag{14}$$

$o(T, L, V)$ is a higher-order term of the Taylor expansion, so the first-order linear Taylor expansion of the magnetic interference is as follows:

$$\Delta H_I = H_I(T + \Delta T, L + \Delta L, V + \Delta V) - H_I(T, L, V) = \frac{\partial H_I}{\partial T}\Delta T + \frac{\partial H_I}{\partial L}\Delta L + \frac{\partial H_I}{\partial V}\Delta V. \tag{15}$$

It can be found that the error of the magnetic interference model is linearly related to the fluxgate measurement errors $\Delta T$, $\Delta L$, and $\Delta V$. The larger the measurement error is, the larger the error of the magnetic interference model of the aircraft platform is. So, the fluxgate magnetometer noise can affect the compensation coefficient solved and reduce the compensation effect.

In order to verify the influence of the fluxgate magnetometer noise on the compensation effect, we compare the influence of fluxgate noise with different amplitude on compensation effect through the Hainan experimental flight data in 2019. The sampling rate is 10 Hz. Add gaussian white noise with an amplitude of 1%, 5%, and 10% of stability amplitude of the fluxgate magnetic field to the fluxgate magnetometer data and the OPM data do not add noise. Then, use the fluxgate magnetometer data and OPM data to solve the compensation coefficients so as to compare the influence of the fluxgate magnetometer noise on compensation effect. The compensation coefficients solved by the fluxgate magnetometer data that added different amplitudes of noise are shown in Table 1.

**Table 1.** The compensation coefficients solved by the fluxgate magnetometer data after adding different amplitudes noise.

| Compensation Coefficients | No White Noise | 1% White Noise | 5% White Noise | 10% White Noise |
|---|---|---|---|---|
| $c_1$ | 2.7063 | 2.7693 | 2.6785 | 2.6026 |
| $c_2$ | 1.6073 | 2.0583 | 1.8476 | 1.6074 |
| $c_3$ | $-0.2453$ | $-0.2356$ | $-0.3369$ | $-0.4062$ |
| $c_4$ | $2.1681 \times 10^{-5}$ | $-5.4560 \times 10^{-5}$ | $-3.6020 \times 10^{-5}$ | $-1.9696 \times 10^{-5}$ |
| $c_5$ | $-4.6649 \times 10^{-6}$ | $-4.5530 \times 10^{-6}$ | $-4.7164 \times 10^{-6}$ | $-9.0702 \times 10^{-6}$ |
| $c_6$ | $-1.0169 \times 10^{-5}$ | $-1.0227 \times 10^{-5}$ | $-7.7326 \times 10^{-6}$ | $-7.7468 \times 10^{-6}$ |
| $c_7$ | $1.6431 \times 10^{-5}$ | $-4.5945 \times 10^{-5}$ | $-3.3852 \times 10^{-5}$ | $-1.7170 \times 10^{-5}$ |
| $c_8$ | $-4.7042 \times 10^{-6}$ | $-3.6484 \times 10^{-6}$ | $-6.4742 \times 10^{-6}$ | $1.0502 \times 10^{-5}$ |
| $c_9$ | $1.469 \times 10^{-4}$ | $6.4670 \times 10^{-5}$ | $2.5494 \times 10^{-5}$ | $1.1084 \times 10^{-5}$ |
| $c_{10}$ | $-9.5252 \times 10^{-4}$ | $-8.3109 \times 10^{-4}$ | $-0.0011$ | $-0.0013$ |
| $c_{11}$ | $-2.5920 \times 10^{-6}$ | $-2.4164 \times 10^{-6}$ | $-9.0160 \times 10^{-7}$ | $-1.1915 \times 10^{-6}$ |
| $c_{12}$ | $1.9710 \times 10^{-6}$ | $2.6852 \times 10^{-6}$ | $3.6735 \times 10^{-6}$ | $9.0144 \times 10^{-6}$ |
| $c_{13}$ | $-2.5627 \times 10^{-6}$ | $-2.8103 \times 10^{-6}$ | $-1.8221 \times 10^{-6}$ | $-1.4755 \times 10^{-6}$ |
| $c_{14}$ | $-9.5580 \times 10^{-4}$ | $-8.3262 \times 10^{-4}$ | $-0.0011$ | $-0.0013$ |
| $c_{15}$ | $2.3457 \times 10^{-5}$ | $2.3677 \times 10^{-5}$ | $1.9844 \times 10^{-5}$ | $7.3923 \times 10^{-6}$ |
| $c_{16}$ | $1.9224 \times 10^{-6}$ | $1.8049 \times 10^{-6}$ | $2.6602 \times 10^{-6}$ | $3.2138 \times 10^{-6}$ |
| $c_{17}$ | $-2.3222 \times 10^{-6}$ | $-3.9508 \times 10^{-6}$ | $-1.7717 \times 10^{-5}$ | $-2.3474 \times 10^{-5}$ |
| $c_{18}$ | $-0.0010$ | $-8.8769 \times 10^{-4}$ | $-0.0011$ | $-0.0014$ |

Figure 1 shows the results before and after compensated by different compensation coefficients. Additionally, (a), (b), (c), and (d) represent the time-domain waveforms of the interfered magnetic field before and after compensation when Gaussian white noise with an amplitude of 0% (noise is not added), 1%, 5%, and 10% of the stability amplitude of the fluxgate magnetic field is added, respectively. In order to evaluate the performance of the four compensation coefficients, the compensation results in Figure 1 are quantitatively analyzed to calculate the standard deviation of the residual interference before and after compensation and the IR, as shown in Table 2.

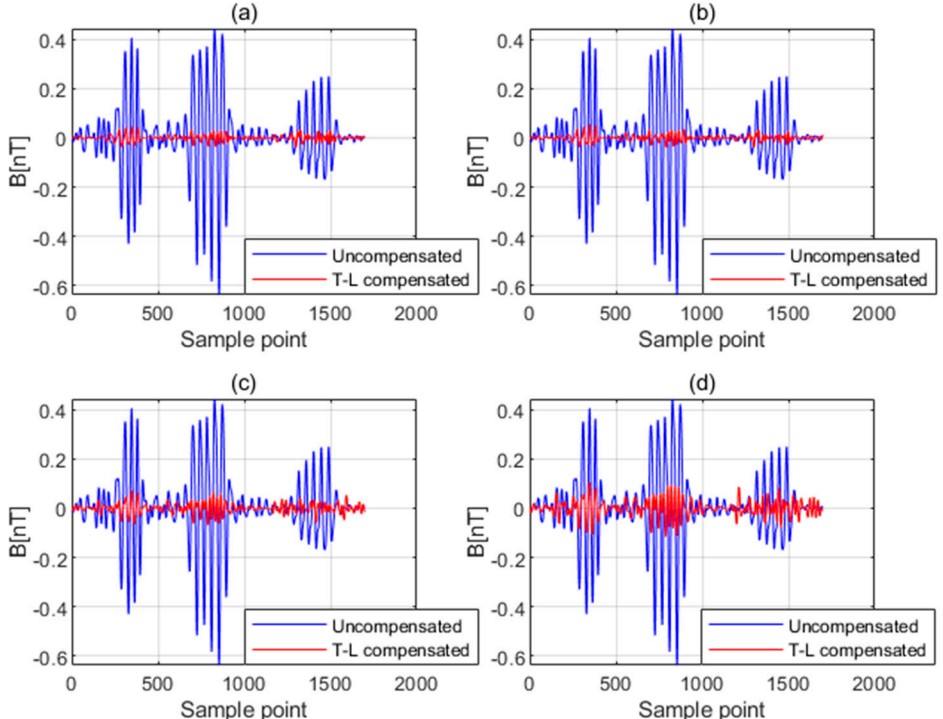

**Figure 1.** The time−domain waveforms of the interfered magnetic field before and after compensation when (**a**) noise is not added, (**b**) the Gaussian white noise with an amplitude of 1% of the stability amplitude of the fluxgate magnetic field is added, (**c**) the Gaussian white noise with an amplitude of 5% of the stability amplitude of the fluxgate magnetic field is added, and (**d**) the Gaussian white noise with an amplitude of 10% of the stability amplitude of the fluxgate magnetic field is added.

**Table 2.** The standard deviation of the residual interference before and after compensated and the IR after different amplitude noises are applied to the fluxgate magnetometer.

| Compensation Coefficients | Before Compensation (rms, pT) | After Compensation (rms, pT) | IR |
|---|---|---|---|
| No white noise | 150.4878 | 11.0523 | 13.6160 |
| 1% white noise | 150.4878 | 12.9478 | 11.6227 |
| 5% white noise | 150.4878 | 24.1168 | 6.2399 |
| 10% white noise | 150.4878 | 35.4036 | 4.2506 |

As can be seen from the compensation results shown in Figure 1, the compensation effect gradually decreases as the amplitude of the added white noise increases. Taking STD as the evaluation index, the larger the amplitude of white noise added, the larger the residual magnetic interference after compensation. It proves that the fluxgate magnetometer noise has a very significant influence on the compensation effect.

To summarize, the larger the noise of the fluxgate magnetometer, the worse the compensation results. The equipment noise is also often unpredictable due to the complex

working conditions and many interference factors. So, it is difficult to improve the compensation effect by adjusting the fluxgate magnetometer. In this article, we propose a new conjunctive compensation method to remove the residual magnetic interference as much as possible.

## 3. Proposed Method

We proposed a new conjunctive compensation method based on INS/GPS and fluxgate magnetometer information. The classical compensation method uses a fluxgate magnetometer to obtain the three components of the geomagnetic field in the body coordinate system to construct the attitude matrix. According to the T-L model, as long as the change in three components of the geomagnetic field in the aircraft coordinate system due to aircraft action can be described, the attitude matrix can be constructed to compensate. So, it is possible to construct an attitude matrix by transforming some auxiliary information that INS/GPS provides.

### 3.1. Compensation Principle

As a magnetic field sensor, the magnetic field measured by the fluxgate is susceptible to interference from external magnetic fields. Meanwhile, the fluxgate itself has the problem of diversionary error, which leads to errors in attitude information obtained by fluxgate measurement during flight. INS is not affected by the magnetic field, but the accuracy is not high. By introducing INS information into the T-L model, attitude information from fluxgate and INS can complement each other to make up for the deficiency of a single sensor.

According to the T-L model, we need to describe the change in three components of the geomagnetic field in the aircraft coordinate system due to aircraft action.

The international geomagnetic reference field (IGRF) is a general international model for describing the earth's main magnetic field through using the longitude, latitude, and altitude information [22]. At present, this model can be used to calculate the seven elements of the geomagnetic field at any point. So, according to the IGRF model, the geomagnetic field parameters (geomagnetic field inclination I, geomagnetic field declination D, and geomagnetic field intensity $H_e$) in the geodetic coordinate system can be calculated by using the longitude, latitude, and altitude information in the aircraft GPS [23]. The aircraft coordinate system can change with the change in roll angle, pitch angle, and heading angle, so it is necessary to convert the geomagnetic field in the geodetic coordinate system to the geomagnetic field in the aircraft coordinate system, as shown in Figure 2.

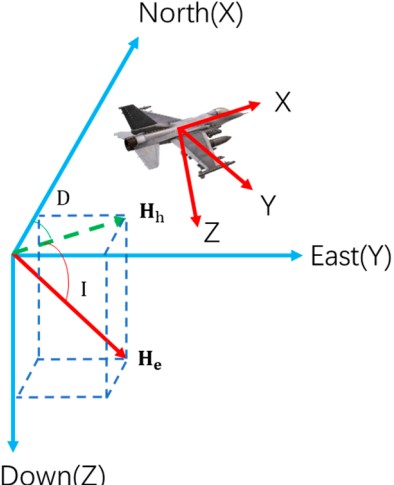

**Figure 2.** Diagram of the coordinate system transformation.

For description, this paper defines the geodetic coordinate system and the aircraft coordinate system as follows:

Geodetic coordinate system: the north-east geodetic coordinate system, with the X-axis pointing to geographic north, the Y-axis pointing to geodetic east, and the Z-axis pointing to the earth's center. $H_e$ denotes the geomagnetic field, $H_h$ denotes the horizontal component of the geomagnetic field, and D and I denote the geomagnetic declination and geomagnetic inclination, respectively.

Aircraft coordinate system: a rectangular coordinate system fixed on an aircraft with the origin at the aircraft's center. The X-axis is parallel to the fuselage axis and points to the flying head. The Y-axis is perpendicular to the longitudinal section of the fuselage and points to the right wings. The Z-axis is perpendicular to the plane where the fuselage is located and points to the bottom of the aircraft. The angle of roll, pitch, and yaw are θ, φ, and ψ, respectively.

According to the IGRF model, the geomagnetic field parameters can be calculated by Equation (16).

$$[H_e,\ I,\ D] = IGRF(time, longitude, latitude, altitude) \tag{16}$$

The XYZ three components of the geomagnetic field in the geodetic coordinate system are shown in the following equations:

$$H_{geox} = \cos I \cos D H_e \tag{17}$$

$$H_{geoy} = \cos I \sin D H_e \tag{18}$$

$$H_{geoz} = \sin I H_e. \tag{19}$$

The XYZ three components of geomagnetic field vector in the aircraft coordinate system are set to ($H_{airx}$, $H_{airy}$, and $H_{airz}$). The XYZ three components of the geomagnetic field vector in the geodetic coordinate system are set to ($H_{geox}$, $H_{geoy}$, and $H_{geoz}$). The angle of roll, pitch, and yaw can directly be obtained by INS. According to the three-dimensional rotation matrix [24], the XYZ three components of the geomagnetic field in the aircraft coordinate system are expressed as follows:

$$\begin{bmatrix} H_{airx} \\ H_{airy} \\ H_{airz} \end{bmatrix} = \phi \begin{bmatrix} H_{geox} \\ H_{geoy} \\ H_{geoz} \end{bmatrix} \tag{20}$$

$$\phi = \begin{bmatrix} 1 & 0 & 0 \\ 0 & \cos\theta & \sin\theta \\ 0 & -\sin\theta & \cos\theta \end{bmatrix} \begin{bmatrix} \cos\varphi & 0 & -\sin\varphi \\ 0 & 1 & 0 \\ \sin\varphi & 0 & \cos\varphi \end{bmatrix} \begin{bmatrix} \cos\psi & \sin\psi & 0 \\ -\sin\psi & \cos\psi & 0 \\ 0 & 0 & 1 \end{bmatrix}. \tag{21}$$

So, we can obtain the change in three components of the geomagnetic field in the aircraft coordinate system due to aircraft action without fluxgate magnetometer data. The T-L compensation model based on fluxgate magnetometer information is expanded and we establish a conjunctive compensation model with 36 coefficients. The expression is as follows:

$$H_I = H_{I_{FLUX}} + H_{I_{IGRF}} \tag{22}$$

$$H_{I_{FLUX}} = \sum_{i=1}^{3} p_i c_i + H_{flux} \sum_{i=1}^{3} \sum_{j=1}^{3} a_{ij} c_i c_j + H_{flux} \sum_{i=1}^{3} \sum_{j=1}^{3} b_{ij} \dot{c}_i c_j \tag{23}$$

$$H_{I_{IGRF}} = \sum_{i=1}^{3} k_i g_i + H_{igrf} \sum_{i=1}^{3} \sum_{j=1}^{3} m_{ij} g_i g_j + H_{igrf} \sum_{i=1}^{3} \sum_{j=1}^{3} n_{ij} \dot{g}_i g_j \tag{24}$$

where $p_i$,$a_{ij}$, and $b_{ij}$ are the compensation coefficients based on fluxgate magnetometer information, and $c_i$ is the cosine of the angle between the geomagnetic field and the three axes of aircraft coordinate system and it can be calculated using the measured values $H_i$, $i = 1, 2, 3$ of the three axes of the fluxgate magnetometer. The expression is (25)and $\dot{c}_i$ is the time derivative of $c_i$.

$$c_i = \frac{H_i}{\sqrt{H_1^2 + H_2^2 + H_3^2}}, i = 1, 2, 3 \tag{25}$$

$H_{flux}$ can be calculated by Equation (26).

$$H_{flux} = \sqrt{H_1^2 + H_2^2 + H_3^2} \tag{26}$$

$k_i$, $m_{ij}$, and $n_{ij}$ are the compensation coefficients based on INS/GPS information, and $g_i$ is the cosine of the angle between the geomagnetic field and the three axes of the aircraft coordinate system calculated by the IGRF and the rotation matrix and it is calculated using the analog values $Hg_i$, $i = 1, 2, 3$ by the IGRF and rotation matrix. The expression is (27) and $\dot{g}_i$ is the time derivative of $g_i$.

$$g_i = \frac{Hg_i}{\sqrt{Hg_1^2 + Hg_2^2 + Hg_3^2}}, i = 1, 2, 3 \tag{27}$$

$H_{igrf}$ can be calculated by Equation (28).

$$H_{igrf} = \sqrt{Hg_1^2 + Hg_2^2 + Hg_3^2} \tag{28}$$

So, the matrix form of Equation (22) can be expressed as follows:

$$H_I = Xw \tag{29}$$

where X is n $\times$ 36 platform attitude matrix and $w$ is 36 $\times$ 1 coefficients matrix.

Then, the least squares solution of the compensation coefficient $w$ can be expressed as follows:

$$w = \left(X^T X\right)^{-1} X^T H_I. \tag{30}$$

*3.2. Compensation Schemes*

This method is the improvement of the T-L compensation model based on fluxgate magnetometer information. Therefore, traditional calibration flights are still applicable to solve the compensation coefficients. The INS/GPS data and the fluxgate magnetometer data are independent, so it can also use only the INS/GPS data to complete the magnetic compensation. After the calibration flight, the calibration flight data are processed, and finally, 36 compensation coefficients of the conjunctive compensation model are obtained. The specific process is as follows:

1. Obtain the original magnetic data from the OPM and preprocess the magnetic data, mainly filtering;
2. Bring the longitude, latitude, and altitude data in GPS into the IGRF and the XYZ three components of the geomagnetic field in the geodetic coordinate system are calculated by IGRF. Bring the angle of roll, pitch, and yaw in INS into the rotation matrix and the XYZ three components of the geomagnetic field in the aircraft coordinate system are calculated by (17)~(19);
3. The same filtering process is used for OPM data, fluxgate magnetometer data, and XYZ three-component data calculated through INS/GPS data, and the conjunctive compensation matrix is constructed in (23) and (24);
4. Calculate the 36 compensation coefficients by the least square algorithm (LS);

5. The modeling interference is calculated by the conjunctive compensation matrix and the compensation coefficients. The modeling interference is subtracted from the filtered OPM data to generate the magnetic field data after compensation.

The data processing flow of the compensation coefficients solution is shown in Figure 3 and the compensation flow is shown in Figure 4.

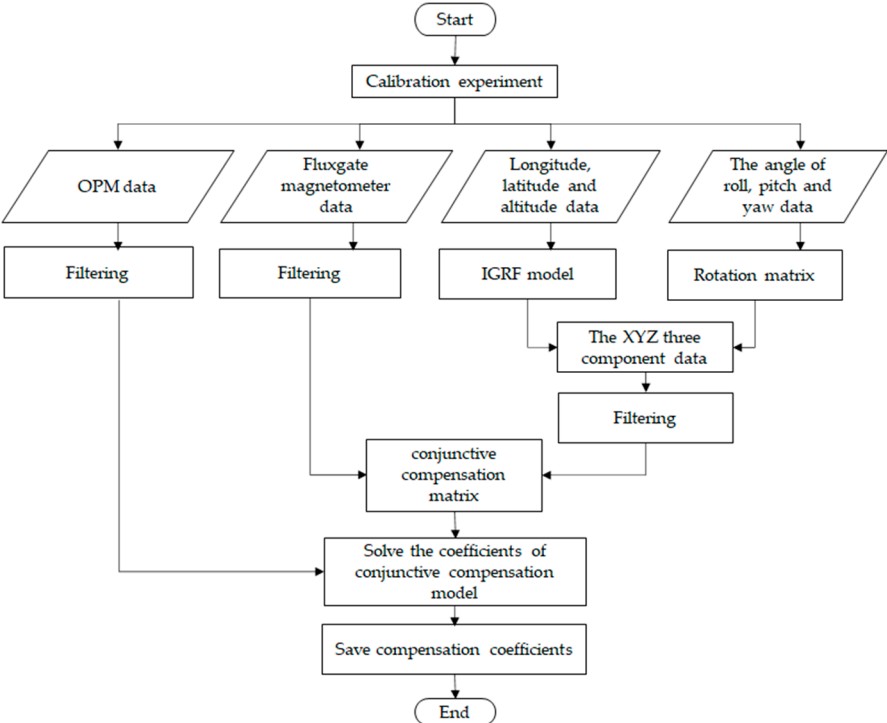

**Figure 3.** Flow chart of compensation coefficient solution.

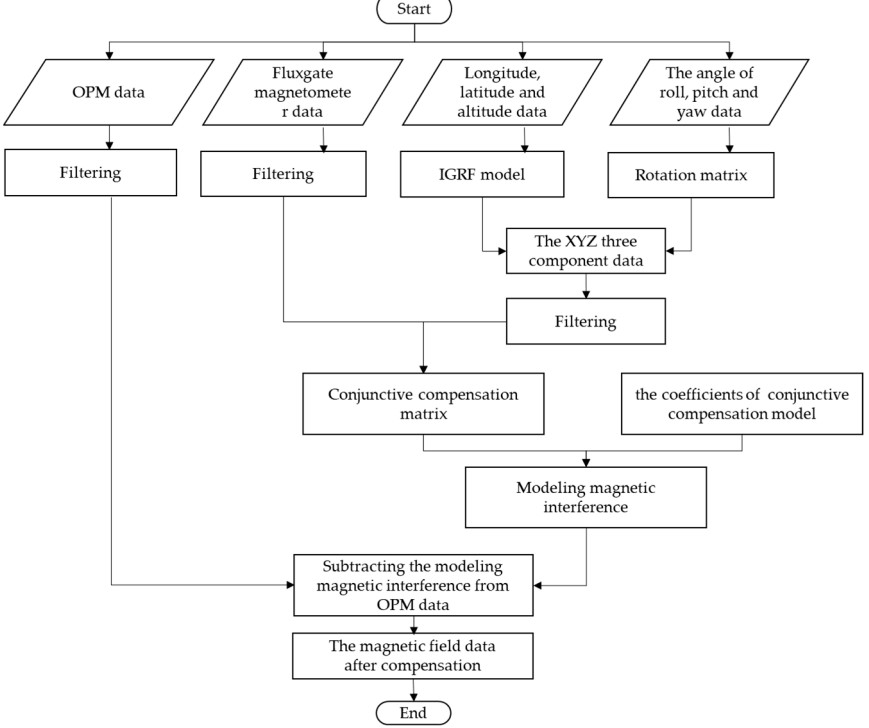

**Figure 4.** Flow chart of magnetic compensation.

## 4. Results

### 4.1. Experimental Result

The flight experiment was conducted in the seas south of Hainan, China, in 12 and 21 July 2022. The experimental region is an open field far away from town and with a stable magnetic field. There are no interference magnetic sources, such as buildings. The aircraft platform is a Y-9 transport aircraft and a connecting rod was installed on the tail cone of the aircraft to reduce the influence of the aircraft's magnetic field interference on the magnetic sensor. The data sampling rate is 10 Hz and filtering bandwidth is 0.04–0.3 Hz. In this experiment, the calibration flight data on July 12 are used as the calibration flight dataset to calculate the compensation coefficients. The calibration flight data on 21 July are used as the verification flight dataset, as shown in the Figure 5. For flight paths (a) on 12 July 2022 and (b) on 21 July 2022, the red part in (a) shows the trajectory of the calibration flight segment on 12 July. The red part in (b) is the trajectory of the calibration flight segment on 21 July. The flight altitude is 3000 m. The planes will perform the rolls ($\pm10°$), pitches ($\pm5°$), and yaws ($\pm5°$) in sequence on each side.

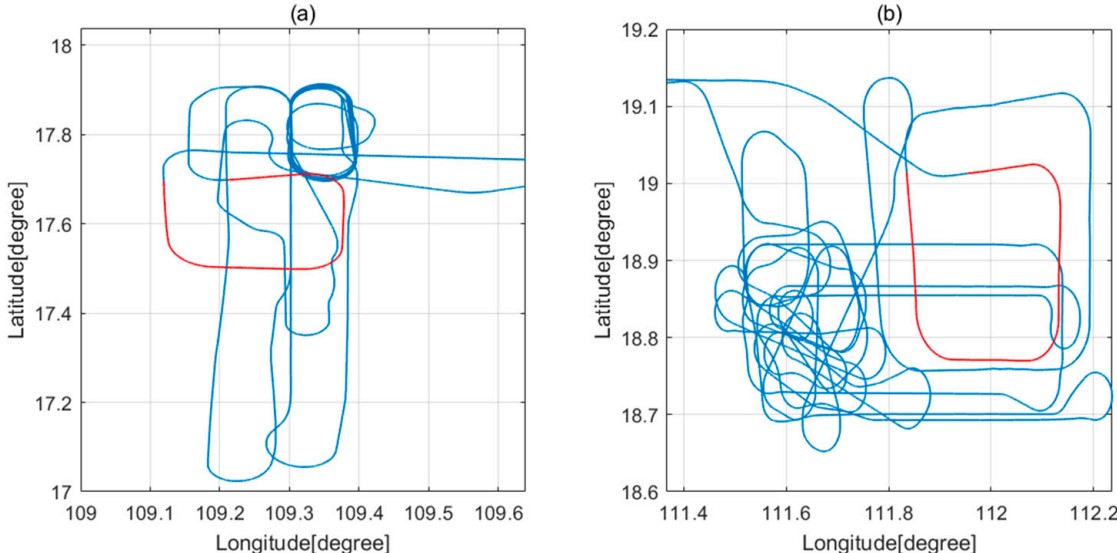

**Figure 5.** Flight paths (**a**) on 12 July 2022 and (**b**) on 21 July 2022.

Using the calibration flight data set, the 18 compensation coefficients only based on the fluxgate magnetometer and only based on INS/GPS information are obtained by the least squares algorithm as shown in Table 3. The 36 compensation coefficients of the conjunctive compensation model are shown in Table 4.

The compensation coefficients in Tables 3 and 4 are substituted into the calibration flight dataset and the verification flight dataset. The calibration flight data before and after compensation and the verification flight data before and after compensation are shown in Figures 6 and 7, respectively. The uncompensated data are shown in (a) and the comparison between compensated data using these three sets of compensation coefficients is shown in (b). Figures 8 and 9 show the magnetic interference power spectral density of the calibration flight data before and after compensation and the verification flight data before and after compensation, respectively. To evaluate the performance of the three compensation coefficients, the STD and IR are calculated as shown in Tables 5 and 6.

**Table 3.** The 18 compensation coefficients are obtained only based on the fluxgate magnetometer and only based on INS/GPS information.

| Compensation Coefficients | Only Based on Fluxgate Magnetometer | Only Based on INS/GPS |
|---|---|---|
| c1 | 18.2005 | 12.3156 |
| c2 | −4.5502 | 17.8194 |
| c3 | 11.9703 | −3.5954 |
| c4 | $-6.7226 \times 10^{-4}$ | $-2.5400 \times 10^{-4}$ |
| c5 | $1.9335 \times 10^{-5}$ | $-1.7171 \times 10^{-5}$ |
| c6 | $-3.1629 \times 10^{-5}$ | $-9.1665 \times 10^{-5}$ |
| c7 | $-2.8612 \times 10^{-4}$ | $-5.6217 \times 10^{-4}$ |
| c8 | $-9.4523 \times 10^{-5}$ | $1.8367 \times 10^{-5}$ |
| c9 | $-4.1923 \times 10^{-4}$ | $-2.2068 \times 10^{-4}$ |
| c10 | −0.0118 | −0.0150 |
| c11 | $-1.2062 \times 10^{-6}$ | $1.5055 \times 10^{-5}$ |
| c12 | $-1.2087 \times 10^{-5}$ | $7.8399 \times 10^{-5}$ |
| c13 | $-5.0812 \times 10^{-5}$ | $-3.3818 \times 10^{-5}$ |
| c14 | −0.0116 | −0.0139 |
| c15 | $-2.6569 \times 10^{-4}$ | $6.1942 \times 10^{-6}$ |
| c16 | $3.3862 \times 10^{-6}$ | $-1.5106 \times 10^{-4}$ |
| c17 | $8.6262 \times 10^{-5}$ | $7.1800 \times 10^{-5}$ |
| c18 | −0.0130 | −0.0136 |

**Table 4.** The 36 compensation coefficients of the conjunctive compensation model.

| Conjunctive Compensation Model Coefficients | Value | Conjunctive Compensation Model Coefficients | Value |
|---|---|---|---|
| c1 | 19.3471 | c19 | −1.8367 |
| c2 | −1.2289 | c20 | 2.2227 |
| c3 | 13.3915 | c21 | 2.2764 |
| c4 | $-5.5459 \times 10^{-4}$ | c22 | $1.5645 \times 10^{-4}$ |
| c5 | $1.9485 \times 10^{-5}$ | c23 | $-1.1847 \times 10^{-4}$ |
| c6 | $-6.2487 \times 10^{-5}$ | c24 | $6.5468 \times 10^{-5}$ |
| c7 | $-2.0969 \times 10^{-4}$ | c25 | $1.5699 \times 10^{-4}$ |
| c8 | $6.2506 \times 10^{-7}$ | c26 | $-4.5371 \times 10^{-6}$ |
| c9 | $-2.1416 \times 10^{-4}$ | c27 | $1.4092 \times 10^{-4}$ |
| c10 | 0.0240 | c28 | −0.0092 |
| c11 | $-2.9659 \times 10^{-5}$ | c29 | $1.7255 \times 10^{-4}$ |
| c12 | $2.4828 \times 10^{-4}$ | c30 | $2.4131 \times 10^{-5}$ |
| c13 | $-5.1210 \times 10^{-5}$ | c31 | $-3.1431 \times 10^{-5}$ |
| c14 | 0.0240 | c32 | −0.0096 |
| c15 | $-1.5170 \times 10^{-4}$ | c33 | $-3.2166 \times 10^{-6}$ |
| c16 | $1.2771 \times 10^{-4}$ | c34 | $2.8985 \times 10^{-5}$ |
| c17 | $9.4952 \times 10^{-5}$ | c35 | $-2.4998 \times 10^{-5}$ |
| c18 | 0.0225 | c36 | −0.0095 |

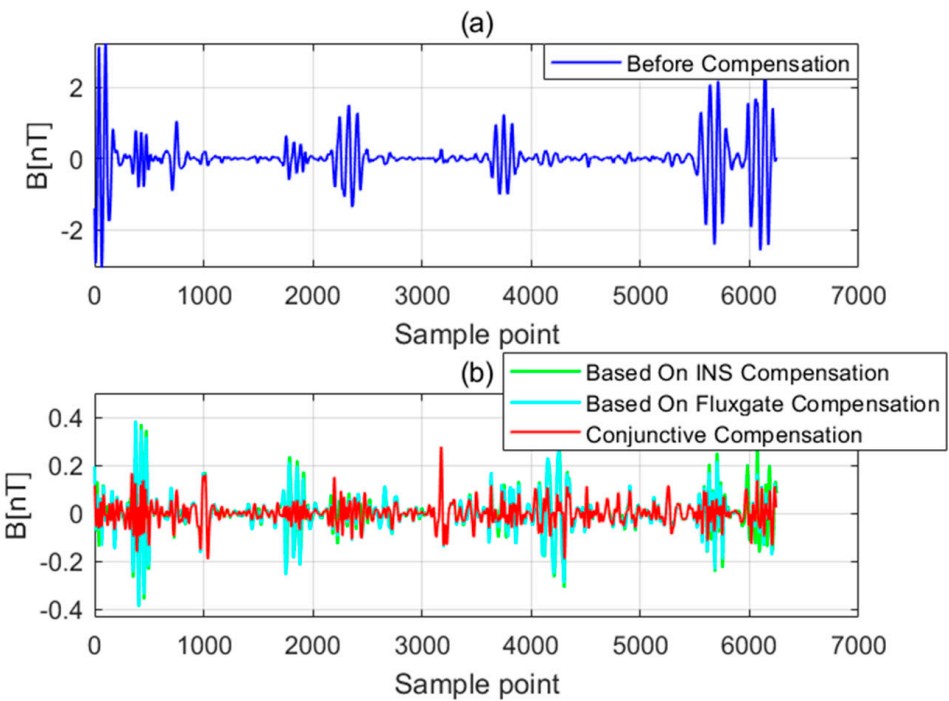

**Figure 6.** The calibration flight data before and after compensated.

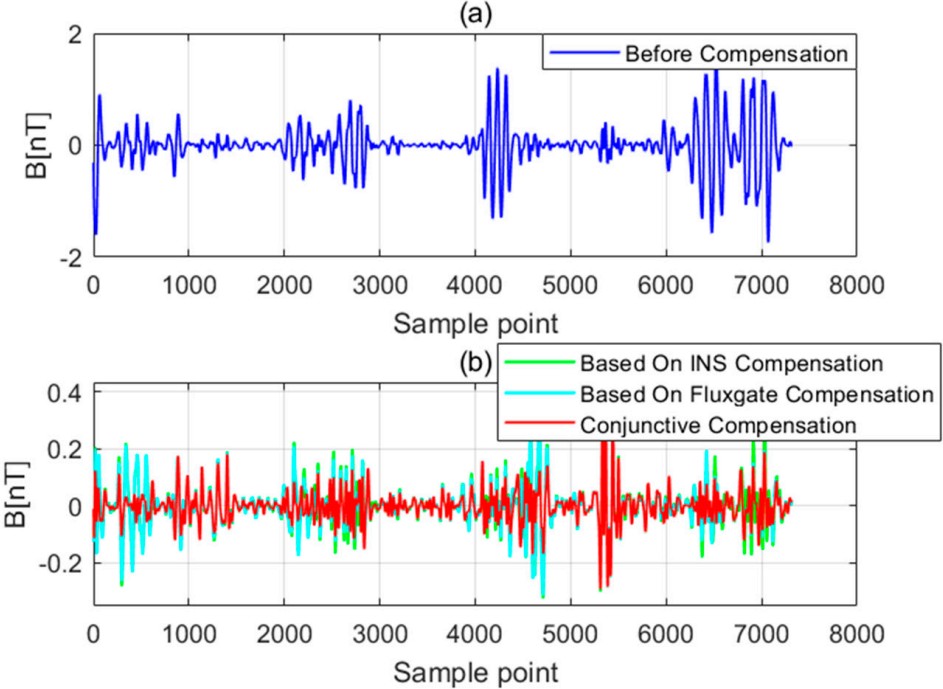

**Figure 7.** The verification flight data before and after compensated.

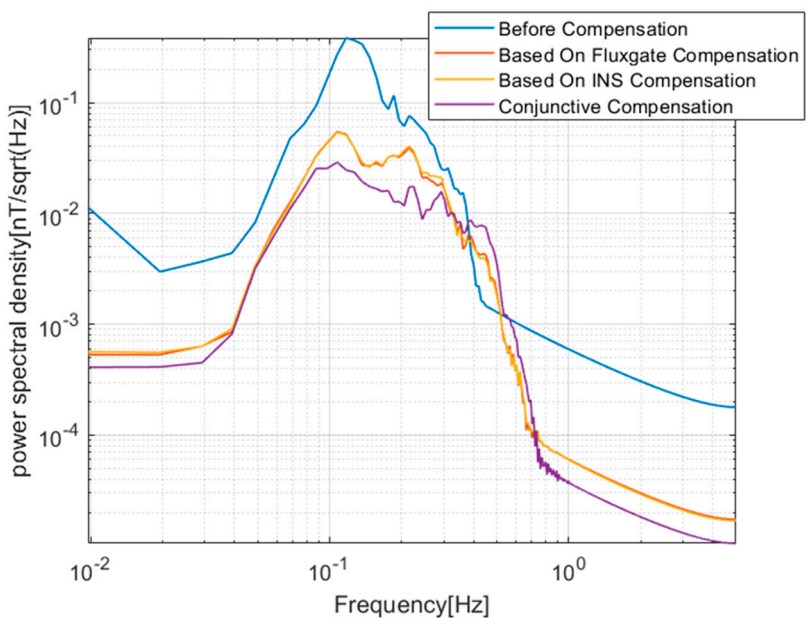

**Figure 8.** The power spectral density of the calibration flight data before and after compensated.

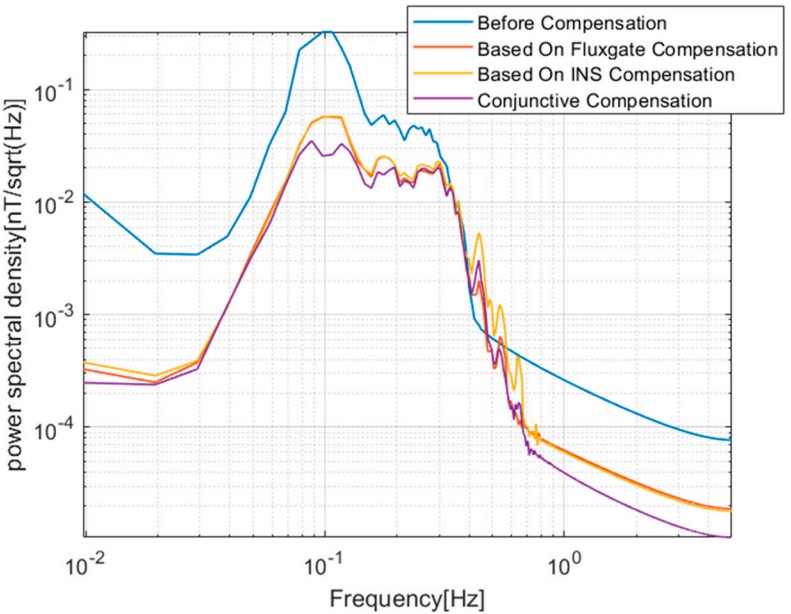

**Figure 9.** The power spectral density of the verification flight data before and after compensated.

**Table 5.** The STD before and after compensated and IR when the calibration flight data are compensated by three methods.

| Calibration Flight Data | Only Based on Fluxgate | Only Based on INS/GPS | Conjunctive Compensation Model |
|---|---|---|---|
| STD before compensation (pT) | 576.1521 | 576.1521 | 576.1521 |
| STD after compensation (pT) | 76.9257 | 80.0745 | 46.2024 |
| IR | 7.4897 | 7.1952 | 12.4702 |

**Table 6.** The STD before and after compensated and IR when the verification flight data are compensated by three methods.

| Verification Flight Data | Only Based on Fluxgate | Only Based on INS/GPS | Conjunctive Compensation Model |
|---|---|---|---|
| STD before compensation (pT) | 386.3243 | 386.3243 | 386.3243 |
| STD after compensation (pT) | 74.6030 | 77.7266 | 56.0597 |
| IR | 5.1784 | 4.9703 | 6.8913 |

*4.2. Result Analysis*

The comparison of the magnetic interference waveforms before and after compensation in Figures 6 and 7 shows that the 18 compensation coefficients only based on INS/GPS information can also effectively compensate for the noise generated by the aircraft. The compensation effect is slightly lower than 18 compensation coefficients only based on fluxgate magnetometer information, but the difference is not obvious. It is considered to cancel the fluxgate magnetometer and use INS/GPS information in some application scenarios of low power consumption and light load.

There are still certain amounts of residual interference in both. It may be caused by fluxgate and INS/GPS measurement error or IGRF modeling error. The conjunctive compensation model can achieve a significant compensation effect through combining the INS information and fluxgate magnetometer information to construct the conjunctive compensation model.

As shown in Figures 8 and 9, in the 0.04–0.3 Hz band of concerning, the magnetic interference spectral density of conjunctive compensation is significantly less than that of the traditional compensation method using only fluxgate information.

As can be seen from Tables 1 and 2, the compensation effect of the conjunctive compensation model is better than that only based on fluxgate magnetometer information 30–60%. The IR is significantly improved and it can be compensated in real time. The experimental data and their processing results show that this method can significantly reduce the residual magnetic interference as an improvement of the traditional compensation method.

**5. Discussion**

In the previous section, the calibration and verification flight results show that the introduction of INS information can effectively improve the compensation effect. In this paper, the introduction of INS information makes the attitude information from the fluxgate and INS complement each other, which reduces the attitude information error calculated by a single sensor and improves the compensation accuracy. However, due to the limitation of time and experimental conditions, this method still has many contents to be further explored. Through the discussion, we hope to inspire readers to think more. For example, on a small UAV, due to the small cabin and high integration of electronic equipment, the electromagnetic environment is more complex. At this time, the fluxgate is subjected to serious equipment magnetic interference. In this case, can the fluxgate still be normally used for aircraft maneuvering compensation? Can INS not affected by the magnetic field be used to replace the fluxgate for maneuvering magnetic interference compensation? This is a question to be explored in the field of UAV magnetic exploration in the future. Another example is that for equipment components, when the magnetic interference reaches certain magnitude, the compensation effect will be lost, which will be helpful for the selection of aeromagnetic system equipment components. In addition, it is found in this paper that magnetic compensation can also be achieved by using only the IGRF model and INS information. Although the effect is not as good as that of the traditional methods, it proves that three-component changes of the geomagnetic field in the body coordinate system

can be estimated by designing flight trajectory and flight actions. Will this help to correct fluxgate errors? These are all possible directions for future research.

### 6. Conclusions

Eliminating the magnetic interference of the aircraft platform is an important technical link of aeromagnetic survey and is a vital part of aeromagnetic measurement. Its compensation effect directly determines the quality of aeromagnetic survey data. The traditional compensation method is based on the T-L model, which establishes a linear relationship between the magnetic interference of the aircraft and the attitude of the aircraft. The compensation coefficients are solved by designing the calibration flight. At present, almost all aeromagnetic systems use the fluxgate magnetometer fixed to the aircraft to realize the attitude measurement of the flight platform. However, in fact, the larger the noise of the fluxgate magnetometer, the worse the compensation results. The equipment noise is also often unpredictable due to the complex working conditions and many interference factors. So, it is difficult to improve the compensation effect by adjusting the fluxgate magnetometer. So, this paper improves on the traditional compensation scheme based on fluxgate information and proposes a new conjunctive compensation method based on INS/GPS and fluxgate magnetometer information. The actual flight experimental data and the processing results show that the proposed method in this paper can significantly improve the quality of aeromagnetic measurement data by 30–60% compared with the traditional compensation method only based on fluxgate information and it is a real-time compensation method. Additionally, the compensation coefficients only based on INS/GPS information can also effectively compensate for the noise generated by the aircraft. The compensation effect is slightly lower than that only based on fluxgate magnetometer information, but the difference is not obvious. It is considered to cancel the fluxgate magnetometer and use INS/GPS information in some application scenarios of low power consumption and light load. In summary, the method proposed in this paper can significantly reduce the residual interference and improve the signal-to-noise ratio of magnetic anomaly signals, which is conducive to the development of subsequent magnetic anomaly detection algorithms.

**Author Contributions:** Conceptualization, B.C. and K.Z.; methodology, B.C.; validation, B.C.; data curation, B.C.; writing—original draft preparation, B.C. and K.Z.; writing—review and editing, B.Y.; project administration, W.Z. All authors have read and agreed to the published version of the manuscript.

**Funding:** This research was funded by the National Key Research and Development Program of China, grant number 2021YFB3900202.

**Institutional Review Board Statement:** Not applicable.

**Informed Consent Statement:** Not applicable.

**Data Availability Statement:** Not applicable.

**Acknowledgments:** The authors would like to thank the editors and reviewers for their efforts to help the publication of this work.

**Conflicts of Interest:** Author declares no conflict of interest.

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
