# Peer review of "The Conjunctive Compensation Method Based on Inertial Navigation System and Fluxgate Magnetometer"

_applsci, doi:10.3390/app13085138_

Round 1
Reviewer 1 Report
Review of “The Conjunctive Compensation Method Based on Inertial Navigation System and Fluxgate Magnetometer”
This paper sets out to describe how using additional information from an inertial navigation system can improve the compensation (removal of noise related to aircraft manoeuvres) of aeromagnetic survey data. This topic is of real interest to many in the aerogeophysical survey sector and the results obtained combining the traditional fluxgate approach and INS data appear convincing.
My main concern with the paper as it stands is that it is not entirely clear how the Conjunctive Compensation Method works. It appears that the method simply adds the solution from the fluxgate to the equivalent solution from the INS (equation 1 and equation 22 outlining the correction from the fluxgate component are identical). The authors state that the resulting 36 coefficients are calculated by a least square method. However, it is not clear that this method will be over-determined, for example coefficient C1 describing fluxgate interaction with the residual fixed field will be utilising the same optically pumped magnetometer (OPM) data as C19 which is describing the interaction with the residual fixed field described by the INS. I do not understand how the values of these two coefficients are distinguished? A better explanation, or clearer statement of how the Conjunctive Compensation Method works is therefore needed. Ultimately I was not sure how I would implement this method (which I would be keen to do). Following on from this I was wondering if as a future development it would be possible to weight the relative impact of the INS/fluxgate? For example you could have a cheap and likely noisy mems INS, or a top of the range navigational grade INS unit with very low noise. Being able to tune the method based on this information would be useful.
My other points are mainly editorial relating to the presentation of the paper, or minor wording choices.
L51 What is meant by “space magnetic total field”? It might be better to say “the magnetic total field”. The term “space” is used several times in this context, and I do not think it is correct, or if it is it is not standard usage.
L117 mentions the “residual magnetic field” as one of the components identified by Tolles and Lawson, also on L161. It may be better to call this the “Permanent magnetic field”, as it matches with the annotation in EQ1 (HIP), and avoids confusion with residual field calculated after subtraction of other corrections.
Equation (8) states HI=HIP+HII+HIE, but HIP isn’t defined in the subsequent lines. On line 161 it is stated that “Where ??? is residual magnetic field”, I think on line 161 HIR should be HIP.
L191 “Error! Reference source not found”. Here and in many other places references to figures are not correct.
L201 and L309 are both labelled as Figure 1. There is an issue with this.
L309 (Fig 1?) I think there should be a line linking the “Fluxgate magnetometer data à filtering” section to the “conjunctive compensation matrix” box in this figure.
Table 3 – there is a horizontal line between c9 and c10 – does this have any meaning/significance? If so put in table caption. If not please remove. Table 5 has a similar horizontal line, not seen in table 6.
Reviewer 2 Report
I recommend processing the following comments:
1. Introduction did not substantiate that this research is currently justified.
2. The majority of literature reviews are out of date.
3. What are the consequences and future directions of research?
4. Add a discussion section to discuss results and findings with other studies.
5. Add more references
6. The English language must be checked
7. Error! Reference source not found, often repeated in the text
Round 2
Reviewer 1 Report
No further commetns
Reviewer 2 Report
Although the literature and the number of references have been changed to some extent, I recommend always focusing on the current and developing the research on the basis of sufficient literature that corresponds to the content of the article.